# Short-Term Outcomes after D2 Gastrectomy with Complete Mesogastric Excision in Patients with Locally Advanced Gastric Cancer: A Systematic Review and Meta-Analysis of High-Quality Studies

**DOI:** 10.3390/cancers16010199

**Published:** 2023-12-31

**Authors:** Stefano Granieri, Annaclara Sileo, Michele Altomare, Simone Frassini, Elson Gjoni, Alessandro Germini, Alessandro Bonomi, Eigo Akimoto, Chun Lam Wong, Christian Cotsoglou

**Affiliations:** 1General Surgery Unit, ASST Brianza—Vimercate Hospital, 20871 Vimercate, Italy; elson.gjoni@asst-brianza.it (E.G.); alessandro.germini@asst-brianza.it (A.G.); 2General Surgery Residency Program, University of Milan, 20122 Milan, Italy; annaclara.sileo@unimi.it (A.S.); alessandro.bonomi@unimi.it (A.B.); 3Trauma Center and Emergency Surgery, ASST Great Metropolitan Hospital Niguarda, 20162 Milan, Italy; michele.altomare@ospedaleniguarda.it; 4General Surgery Residency Program, University of Pavia, 27100 Pavia, Italy; simone.frassini01@universitadipavia.it; 5Department of General Surgery, Juntendo University Nerima Hospital, Tokyo 177-8521, Japan; e.akimoto@juntendo-nerima.jp; 6Ruttonjee & Tang Siu Kin Hospital, Hong Kong, China; wcl951@ha.org.hk

**Keywords:** gastric cancer, D2 lymphadenectomy, complete mesogastric excision, systematic review, meta-analysis

## Abstract

**Simple Summary:**

Local recurrence is a significant issue for advanced gastric cancer patients. Complete mesogastric excision (CME) has been advocated to enhance lymph node (LN) retrieval and reduce recurrence rates. A systematic review of the literature was conducted according to the Cochrane recommendations, and meta-analyses of means and binary outcomes were developed. The number of lymph nodes retrieved was the primary endpoint, with other postoperative outcomes as secondary. Thirteen studies were included, showing that the mean number of harvested LNs was significantly higher among patients undergoing CME. CME patients also had significantly lower intraoperative blood loss, a shorter length of stay, and a shorter operative time. Radical gastrectomy with CME may provide a safe and more radical lymphadenectomy. Long-term outcomes and the applicability of this technique in the West are still to be proven.

**Abstract:**

Complete mesogastric excision (CME) has been advocated to allow for a more extensive retrieval of lymph nodes, as well as lowering loco-regional recurrence rates. This study aims to analyze the short-term outcomes of D2 radical gastrectomy with CME compared to standard D2 gastrectomy. A systematic review of the literature was conducted according to the Cochrane recommendations until 2 July 2023 (PROSPERO ID: CRD42023443361). The primary outcome, expressed as mean difference (MD) and 95% confidence intervals (CI), was the number of harvested lymph nodes (LNs). Meta-analyses of means and binary outcomes were developed using random effects models to assess heterogeneity. The risk of bias in included studies was assessed with the RoB 2 and ROBINS-I tools. There were 13 studies involving 2009 patients that were included, revealing a significantly higher mean number of harvested LNs in the CME group (MD: 2.55; 95% CI: 0.25–4.86; 95%; *p* = 0.033). The CME group also experienced significantly lower intraoperative blood loss, a lower length of stay, and a shorter operative time. Three studies showed a serious risk of bias, and between-study heterogeneity was mostly moderate or high. Radical gastrectomy with CME may offer a safe and more radical lymphadenectomy, but long-term outcomes and the applicability of this technique in the West are still to be proven.

## 1. Introduction

Gastric cancer (GC) is the fifth most frequent tumor worldwide with a particularly remarkable incidence in East Asia [1]. Despite the crucial role of neoadjuvant chemotherapy in treating advanced-stage diseases, surgery represents the primary curative option [2]. Radical gastrectomy combined with D2 lymphadenectomy represents the standard of care in both Western and Eastern countries, however, recent series demonstrated a high incidence of loco-regional recurrence after surgery, reaching 26.1% and 30.4% at two and five years, respectively [3]. This represents a leading cause of the dismal prognosis of GC patients.

From an embryological point of view, the mesogastrium develops from the rotation, elongation, and interposition of the gastric mesentery. This latter element originally consists of the dorsal mesogastrium (DM) and the ventral mesogastrium (VM). During stomach rotation, the DM constitutes the greater omentum, and the VM becomes the lesser omentum [4,5]. This developmental process involves the embedding of parenchymal organs and vascular structures [6], resulting in avascular interfaces between the mesogastrium and adjacent structures. Thus, the fusion (loose connective tissue arising from the contact with the mesothelium of the serosa of contiguous organs) and the investing fascia (the thin elastic innervated connective tissue that encloses the embedding parenchymal organs and vessels) [7] recreate the equivalent of the mesorectal “holy plane” [8].

For a detailed description of standard D2 radical gastrectomy, we recommend referring to previously published papers [9,10,11,12].

The surgical technique of D2 radical gastrectomy + CME begins with dissecting the gastrocolic ligament until exposing the right gastroepiploic vein (RGEV), artery (RGEA), and the pancreas tail. The second step involves separating the perigastric fascia along the gastroduodenal artery and the common hepatic artery until the right gastric artery is exposed. The hepatoduodenal ligament is dissected, the portal vein exposed, and the residual perigastric fascia is dissected from the gastropancreatic layer toward the right. Subsequently, the splenic recess is dissected to expose the posterior edge of the middle–upper spleen along the left Gerota’s fascia, the right crus of the diaphragm, the esophageal hiatus, and skeletonizing the celiac trunk along with the left gastric artery. The last step consists of the dissection of the surrounding mesenteries and ligaments [13,14,15].

Total mesorectal excision and complete mesocolic excision revolutionized surgical care, reducing the risk of local recurrence through negative margins achievement along embryologic planes [16]. Based on similar anatomical and embryological principles, complete mesogastric excision (CME) has gained consensus. The existence of “Metastasis V”, a novel metastatic pathway defined by cancer cells spread through the mesogastrium, is considered an additional risk factor for local recurrence [17,18].

This study aims to analyze the short-term outcomes of D2 radical gastrectomy with complete mesogastric excision compared to standard radical D2 gastrectomy.

## 2. Materials and Methods

The protocol for the present systematic review was registered on PROSPERO with the ID: CRD42023443361. The review adheres to the Preferred Reporting Items for Systematic Reviews and Meta-Analyses (PRISMA) and the Assessing the Methodological Quality of Systematic Reviews (AMSTAR) guidelines [19,20].

The PubMed, Scopus, and Cochrane Library databases were screened without time restrictions up until 2 July 2023, incorporating all MeSH terms. The full search queries are available in the Appendix A section.

Articles without freely accessible full text were sought through the University of Milan digital library and direct contact with authors. A hand-search of references in included studies and previous reviews on the topic was also performed to include additional relevant studies according to the selection criteria. If articles found through the hand-search were not available in Western databases, searches were extended to the China Knowledge Resource Integrated Database (EA, CLW). The literature search was independently conducted by two investigators (SG, AS).

### 2.1. Inclusion Criteria

A PICOS framework (population, intervention, comparator, outcome, study design) was employed to define study eligibility. Specifically:-Population (P): patients suffering from non-metastatic gastric cancer;-Intervention (I): D2 gastrectomy with complete mesogastric excision;-Comparison (C): conventional D2 gastrectomy;-Outcomes (O): intraoperative and short-term postoperative outcomes (2.6 Primary and secondary outcomes);-Study design (S): all study designs.

Studies with insufficient reporting of the PICOS criteria were excluded.

Only studies comparing D2 radical gastrectomy + CME (I) with standard D2 radical gastrectomy (C) were included.

D2 lymphadenectomy was defined according to the Japanese Gastric Cancer treatment guidelines [21].

### 2.2. Exclusion Criteria

Studies meeting the following criteria were deemed non-eligible:(1)Studies including patients suffering from gastric neoplasms different from adenocarcinoma;(2)Studies including patients suffering from esophagogastric junction without separate outcome data;(3)Studies reporting overlapping series;(4)Case reports, editorials, abstracts, unpublished studies, book chapters, and commentaries;(5)Previously published reviews.

One author (SF) assessed the methodological quality of each retrospective comparative study using the validated Newcastle–Ottawa Scale (NOS) [22].

### 2.3. Systematic Review Process and Data Extraction

The Rayyan web application was used to identify and remove duplicates among identified records [23]. Initially, 782 articles were identified, and after the exclusion of duplicates, two independent reviewers (AS, MA) screened titles and abstracts of 667 records. An a priori-developed screening form was created to guide study selection. Investigators were blinded to each other’s decisions. Disagreements were solved by a third party (CC) who supervised the systematic review process.

Eighteen articles were assessed for eligibility, and finally, thirteen studies fulfilling all inclusion criteria were selected for qualitative and quantitative analysis. The flowchart depicting the overall review process according to PRISMA is reported in Figure 1.

Data extraction was independently performed by three authors (SG, AS, MA). Information regarding study design and methodology, participant demographics and baseline characteristics, operative characteristics, and pathological and postoperative outcomes were compiled in a computerized spreadsheet (Microsoft Excel 2016; Microsoft Corporation, Redmond, WA, USA).

Disagreements were resolved through discussion with the assistance of two additional investigators (AG, EG).

### 2.4. Assessment of the Risk of Bias

The risk of bias for individual studies was evaluated using the RoB 2 and the ROBINS-I tools (for randomized and non-randomized studies, respectively) provided by the Cochrane Collaboration [29,30], conducted by one investigator (SF). The following domains were explored: (1) bias arising from the randomization process; (2) bias due to deviations from intended interventions; (3) bias due to missing outcome data; (4) bias in measurement of the outcome; and (5) bias in selection of the reported results.

Data were collected according to the methodology proposed by Higgins [29] in a computerized spreadsheet. Bar and traffic light plots were generated to graphically display the results of the risk of bias assessment.

### 2.5. Primary and Secondary Outcomes

The primary outcome was the number of harvested lymph nodes. Secondary outcomes included operative time, intraoperative bleeding, time to first postoperative flatus, time to first liquid intake, hospital stay, and postoperative complications.

### 2.6. Statistical Analysis

The primary outcome measure was expressed as mean difference (MD) and 95% confidence intervals (CI). Means were retrieved from each manuscript. Whenever not overtly reported, they were computed from medians, ranges, interquartile ranges (IQR), and sample sizes according to Wan’s method [31].

Secondary outcome measures were reported as MD or odds ratio (OR) with 95% CI as appropriate. Meta-analyses of means and binary outcomes were developed.

Random effects models based on the inverse variance method and DerSimonian–Laird estimator for τ^2^ were built to assess the impact of heterogeneity on results. Between-studies heterogeneity was quantified by I^2^ statistic and Cochran’s Q test. Cut-off values of 25%, 50%, and 75% were considered low, moderate, and high, respectively [32], and 95% of prediction intervals (PI) were provided as well. Sensitivity analysis was conducted to explore the presence of outliers; influential cases were detected through the leave-one-out method and graphical display of study heterogeneity (GOSH) analysis. Forest plots were developed to display the results graphically.

Subgroup analysis and mixed-effect metaregression models were realized to detect eventual further sources of heterogeneity and to identify specific predictors of effect size.

Funnel plots were developed to explore publication bias, and Egger’s test of the intercept was used to quantify funnel plots’ asymmetry. Duval and Tweedie’s trim-and-fill method was adopted to estimate and adjust the number and outcomes of missing studies each time Egger’s test demonstrated significant asymmetry.

Statistical analysis was conducted with R statistical software (The Comprehensive R Archive Network—CRAN, ver. 4.0.0 x64) [33], using “meta”, “metafor”, “robvis”, and “dmetar” packages [34,35,36,37].

## 3. Results

### 3.1. Descriptive Noncomparative Analysis of Included Studies

After the literature search, 13 studies [38,39,40,41,42,43,44,45,46,47,48,49,50] were included in the qualitative and quantitative analysis. Details of the studies originally sought for retrieval, but eventually excluded, are reported in the PRISMA flow diagram.

In total, 2009 patients were included in the meta-analysis. Of these, 1125 underwent CME, and 984 underwent standard D2 lymphadenectomy. All studies were conducted in China. Among the included studies, five were retrospective, eight were retrospective with random selection, and one was a randomized control trial (RCT). Twelve studies specified the type of gastrectomy: in seven studies [39,41,42,44,45,47,48], all patients underwent total gastrectomy, and in three studies [38,43,49], all patients received subtotal distal gastrectomy. Further details are reported in Table 1.

### 3.2. Primary Outcome

Eleven studies reported the mean number of lymph nodes harvested [39,40,41,42,43,44,45,46,47,48,49,50]. The mean number of LNs harvested was significantly higher among patients receiving CME compared to standard D2 lymphadenectomy (MD: 2.55; 95% CI: 0.25–4.86; 95% PI: −4.68–9.79; *p* = 0.033). Between-studies heterogeneity was significantly high (τ = 3.03; I^2^ = 88.1%, Cochrane Q *p* < 0.0001). The results are graphically reported in Figure 2.

### 3.3. Secondary Outcomes

Thirteen studies reported data regarding intraoperative blood loss [38,39,40,41,42,43,44,45,46,47,48,49,50]. Patients undergoing CME had a significantly lower intraoperative blood loss compared to standard D2 (MD: −35.45 mL; 95% CI −47.48–−23.41; 95% PI: −67.15–−3.75; *p* < 0.0001). High heterogeneity was detected (τ = 13.3; I^2^ = 91.3%, Cochrane Q *p* < 0.0001).

Thirteen studies reported data regarding operative time [38,39,40,41,42,43,44,45,46,47,48,49,50]. No difference between patients undergoing CME and standard D2 was highlighted (MD: −8.71 min; 95% CI −20.5–3.07; 95% PI: −53–35.57; *p* = 0.13). High heterogeneity was detected (τ = 19.38; I^2^ = 95%, Cochrane Q *p* < 0.0001).

Nine studies reported data regarding time to first flatus [38,39,40,41,42,43,45,46,48,50]. Patients undergoing CME had a significantly faster gas and stool passage compared to standard D2 (MD: −0.28 days; 95% CI −0.47–−0.09; 95% PI: −0.72–0.16; *p* = 0.0093). Moderate heterogeneity was detected (τ = 0.17; I^2^ = 61%, Cochrane Q *p* = 0.0086).

Eight studies reported data regarding time to liquid intake [38,39,40,42,43,45,47,48,50]. No difference between patients undergoing CME and standard D2 were highlighted (MD: −0.25 days; 95% CI −0.54–0.04; 95% PI: −0.93–0.43; *p* = 0.079). High heterogeneity was detected (τ = 0.25; I^2^ = 88.1%, Cochrane Q *p* < 0.0001).

Ten studies reported data regarding length of hospital stay (LOS) [38,39,40,41,42,43,45,46,47,48,50]. Patients undergoing CME had a significantly lower LOS compared to standard D2 (MD: −1.03 days; 95% CI −1.79–−0.23; 95% PI: −3.37–1.32; *p* = 0.015). High heterogeneity was detected (τ = 0.95; I^2^ = 84.7%, Cochrane Q *p* < 0.0001).

Twelve studies reported data regarding postoperative complications [38,39,40,41,42,43,45,46,47,48,49,50]. No differences were pointed out between CME and standard D2 patients (OR: 0.76; 95% CI 0.54–1.07; 95% PI: 0.34–1.66; *p* = 0.11). The heterogeneity was low (τ = 0.3; I^2^ = 13.4%, Cochrane Q *p* = 0.31) (Figure 3A).

### 3.4. Sensitivity Analysis

No outliers or influential studies were detected for the primary outcome.

In regards to intraoperative bleeding, outliers, influential, and GOSH analysis identified the studies by Li 2015, Yu, and Li 2023 as responsible for heterogeneity. After their exclusion, a significant blood loss reduction was confirmed for patients receiving CME (MD: −23.79 mL; 95% CI −26.3–−21.29; 95% PI: −26.35–−21.25; *p* < 0.0001). Heterogeneity dropped to zero (τ = 0; I^2^ = 0%, Cochrane Q *p* < 0.48) (Figure 3B).

Regarding operative time, after sensitivity analysis, the studies by Ma, Yu, Zheng, Xie, and Li were identified as overtly contributing to heterogeneity. After excluding them, patients undergoing CME showed a significantly shorter operative time compared to standard D2 LND (MD: −16.11 min; 95% CI −17.74–−14.48; 95% PI: −17.79; −14.42; *p* < 0.0001). Heterogeneity dropped to zero (τ = 0; I^2^ = 0%, Cochrane Q *p* = 0.98) (Figure 3C).

Regarding time to first flatus, after removing the studies by Dang and Zheng, a significantly faster gas and stool passage was confirmed for CME patients (MD: −0.30 days; 95% CI −0. 42–−0.19; 95% PI: −0.46–−0.14; *p* = 0.0007). Heterogeneity dropped to low (τ = 0.04; I^2^ = 6.7%, Cochrane Q *p* = 0.38) (Figure 3D).

Regarding the time to first liquid intake, after removing the studies by Yu and Zheng, no significant difference between the two groups was confirmed (MD: −0.21 days; 95% CI −0.43–0.01; 95% PI: −0.86–0.44; *p* = 0.06). Heterogeneity remained high (τ = 0.21; I^2^ = 85.3%, Cochrane Q *p* < 0.0001) (Figure 3E).

A significant reduction in the length of hospital stay was confirmed after sensitivity analysis for patients undergoing CME (MD: −0.55 days; 95% CI −0.72–−0.38; 95% PI: −0.73–−0.37; *p* = 0.0001). Heterogeneity dropped to zero (τ = 0; I^2^ = 0%, Cochrane Q *p* = 0.94) (Figure 3F).

### 3.5. Subgroup Analysis and Metaregression of Primary Endpoint

The type of surgical approach (open vs. minimally invasive), type of gastrectomy (total vs. distal), study design, and risk of bias were defined as moderators for subgroup analysis. Year of publication, years of enrollment, and total number of patients were defined as moderators for metaregression analysis.

The results are displayed in Table 2.

Subgroup analysis highlighted significant between-group heterogeneity for study design (*p* = 0.04) and type of gastrectomy (*p* = 0.039). The year of publication was significantly related to the effect size (test of moderators *p* < 0.0001). Furthermore, it significantly accounted for most of the between-study heterogeneity (I^2^ = 64.7%; Cochran’s Q *p* = 0.0039), and 78.6% of the difference in true effect sizes (R^2^) can be explained by this moderator.

Funnel plots of subgroup analysis and bubble plots of metaregression are available in the Appendix A.

### 3.6. Quality of the Studies and Risk of Bias Assessment

Figure 4 summarizes the risk of bias evaluation according to the RoB 2 and the ROBINS-I tools for randomized and non-randomized studies, respectively. Three studies were burdened by serious risk of bias. Among non-randomized studies, bias due to confounding was the domain accounting for most of the risk of bias observed. More detailed information is displayed in the traffic light plot reported in the Appendix A.

### 3.7. Assessment of Publication Bias

According to the funnel plot (Figure 5) and Egger’s test (*p* = 0.052), the possible presence of publication bias for the primary endpoint cannot be ruled out. However, this evaluation is doubtful due to the small number of studies included in the analysis.

Egger’s test of the intercept and the assessment of funnel plots detected significant publication bias for intraoperative blood loss (*p* = 0.049).

No publication bias was detected for operative time (*p* = 0.71), time to first flatus (*p* = 0.65) time to first liquid intake (*p* = 0.086), and length of hospital stay (*p* = 0.71).

Funnel plots of publication bias for all secondary endpoints are reported in the Appendix A.

## 4. Discussion

Radical gastrectomy with lymphadenectomy remains the cornerstone of the treatment of patients suffering from locally advanced gastric cancer. Unfortunately, up to 80% of patients experience loco-regional recurrence after surgery, a major factor in their dismal prognoses [51]. The membrane-based approach characterizing CME exploits the anatomical and embryological properties of the mesogastrium, which has been compared to the “holy plane” of the total mesorectal excision (TME).

The present study demonstrated that radical gastrectomy with D2 lymphadenectomy with complete mesogastric excision is associated with a significantly increased number of harvested lymph nodes compared to standard D2 lymphadenectomy. However, high between-studies heterogeneity was observed, and sensitivity analysis identified the year of publication, study design, and type of gastrectomy as the principal moderators responsible for this heterogeneity.

Regarding secondary endpoints, CME was linked to a significantly lower intraoperative blood loss, operative time, time to first flatus, and length of hospital stay compared to standard D2 LND. No difference in the time to liquid intake and postoperative complications was noted. The exclusion of outliers/influential studies explained the heterogeneity found for each moderator.

Despite the encouraging results, the difference between statistical and clinical significance should be kept in mind. Both techniques ensure a high nodal retrieval, and harvesting an average of 2.5 more lymph nodes with CME might not be enough to reduce the risk of local recurrence in advanced gastric cancer patients. Similarly, 23.8 mL less intraoperative blood loss, 16 min shorter operative time, and 0.3 days shorter time to first flatus might not substantially impact the postoperative course. Although the difference was significant, the length of hospital stay for CME patients was only half a day shorter compared to standard D2 gastrectomy, and no differences were found regarding postoperative complications. The better results we obtained, even though perhaps not clinically significant, may open the debate on the ideal en bloc lymph node dissection in surgical oncology. Similarly to TME in rectal cancer and complete mesocolic excision in right-sided colon cancer surgery, this embryology-based technique could explain the slightly better perioperative outcomes due to the dissection through avascular planes [8,52]. These findings reinforce the concept [13] that CME should not be seen as an extension of D2 radical gastrectomy, but rather as its ideal form as suggested by Shinohara et al. [13]. In a previous study by the same group in 2017 [18], the median number of lymph nodes harvested with D2 + CME was almost overlapping with that of the Dutch trial [53]. According to the authors, CME is not expected to increase the number of harvested lymph nodes or contribute to better survival outcomes compared to high-quality D2 gastrectomy. Nonetheless, the membrane approach underlying CME—involving the dissection of adipose tissue along with the lymph nodes while preserving the investing fascia—may aid in preventing the spread of tumor cells through the so-called fifth route of metastasis [5].

The present study represents the most updated systematic review and meta-analysis on complete mesogastric excision. In comparison to the study published nearly 2 years ago by Meng et al. [54], it exclusively includes high-quality studies, encompassing the only RCT currently available on the subject. Moreover, it adheres to the rigorous methodology recommended by the most recent guidelines and employs advanced statistical techniques to identify sources of heterogeneity.

### Limitations

Readers are urged to interpret our results cautiously due to the noteworthy limitations of this study. Indeed, it should be noted that excluding one RCT, all included studies were retrospective, with seven out of thirteen based on random selection.

All the studies included came from China, and various groups have proposed over the years different interpretations of the embryological development of the mesogastrium, leading to differences in surgical approach. The predominant representation of the mesogastrium is the so-called table model proposed by Xie et al. in 2015, outlining a systematic surgical approach based on five “quality control” parameters to assess the integrity of the mesogastrium itself [14]. Radical D2 gastrectomy with CME is not a straightforward procedure, especially when dissecting the suprapancreatic area. It requires advanced laparoscopic surgical skills with a steep learning curve [14,15]. Indeed, most of the procedures in each study were performed by a narrow cluster of extremely skilled laparoscopic surgeons. Given these considerations and the well-known disparities between Eastern and Western populations, the applicability of our findings in a Western setting may be limited. Western patients are often obese, present with advanced-stage diseases, and frequently require neoadjuvant chemotherapy. Many studies in the present meta-analysis excluded patients undergoing neoadjuvant treatment (chemotherapy/chemoradiation) [41,44,48,49,50] and for most of them, no information was reported. This aspect should be considered when evaluating the applicability of this surgical technique in the Western population, as most European and US gastric cancer treatment guidelines recommend neoadjuvant treatment for advanced-stage disease.

Lastly, the present meta-analysis focused on short-term postoperative outcomes intentionally considering the lack of survival data, and as a result most of the included studies did not conduct a proper survival analysis. Many studies reported only survival rates and proportions at one- and/or three-years, with only the study by Li et al. [50] presenting the results of the Kaplan–Meier analysis. In this latter study, despite finding no difference being between D2 and D2 + CME in terms of OS and DFS, patients undergoing complete mesogastric excision experienced significantly fewer local recurrences.

Given the lack of appropriate follow-up with censored data and the aforementioned limitations, in our opinion, exploring long-term outcomes was not deemed worthwhile. Consequently, the long-term results of ongoing studies, especially the RCT led by Gong (NCT01978444) and the prospective observational study led by Yanchang (ChiCTR2200058556), are eagerly awaited.

## 5. Conclusions

Radical gastrectomy with D2 lymphadenectomy and CME emerges as a promising approach for locally advanced gastric cancer. The membrane-based CME technique, akin to the principles of total mesorectal excision, showcases advantages in terms of increased lymph node retrieval and reduced intraoperative blood loss, operative time, and hospital stay compared to standard D2 lymphadenectomy. However, caution is warranted in the interpretation of these findings, as the observed differences may not necessarily translate into clinically significant outcomes. Long-term outcomes and the applicability of such a demanding technique in the West are still unproven.

## Figures and Tables

**Figure 1 cancers-16-00199-f001:**
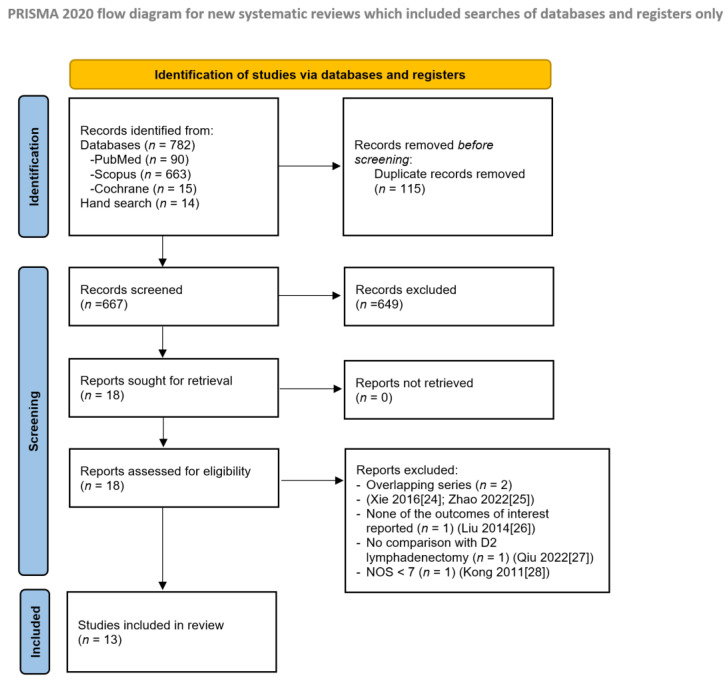
PRISMA flow diagram [24,25,26,27,28].

**Figure 2 cancers-16-00199-f002:**
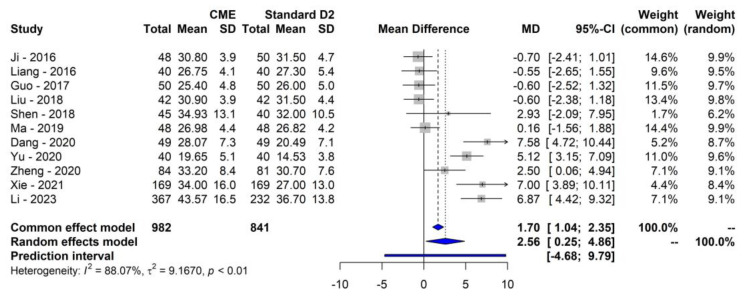
Forest plot of mean difference for LNs harvested [39,40,41,42,43,45,46,47,48,49,50].

**Figure 3 cancers-16-00199-f003:**
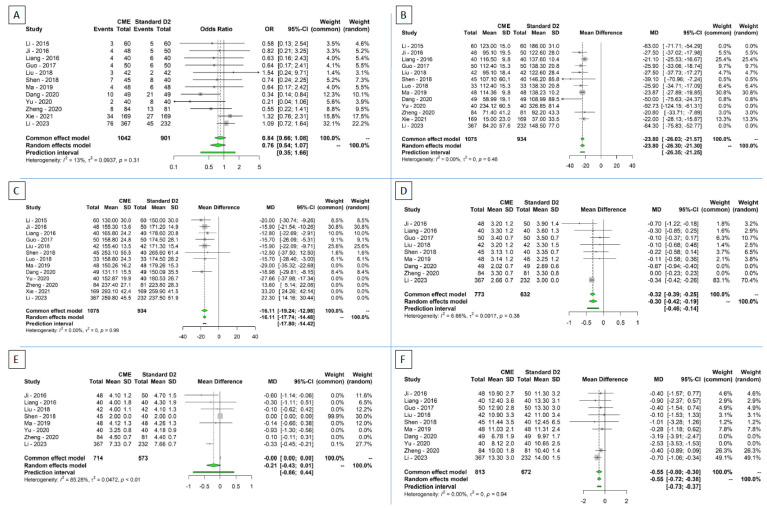
Secondary outcomes. Forest plots for (**A**) postoperative complications, (**B**) intraoperative bleeding, (**C**) operative time, (**D**) time to first flatus, (**E**) time to first liquid intake; (**F**) length of hospital stay [38,39,40,41,42,43,44,45,46,47,48,49,50].

**Figure 4 cancers-16-00199-f004:**
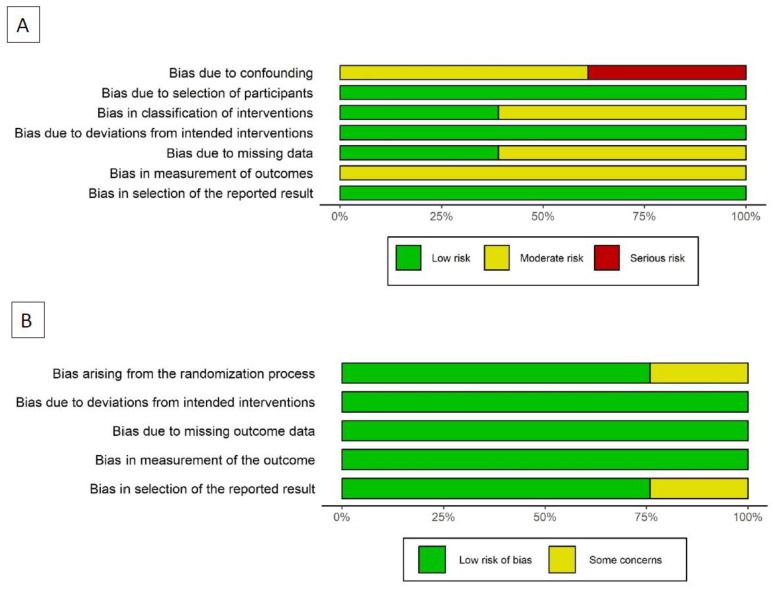
Risk of bias assessment through barplot: (**A**) Non-randomized studies; (**B**) Randomized studies.

**Figure 5 cancers-16-00199-f005:**
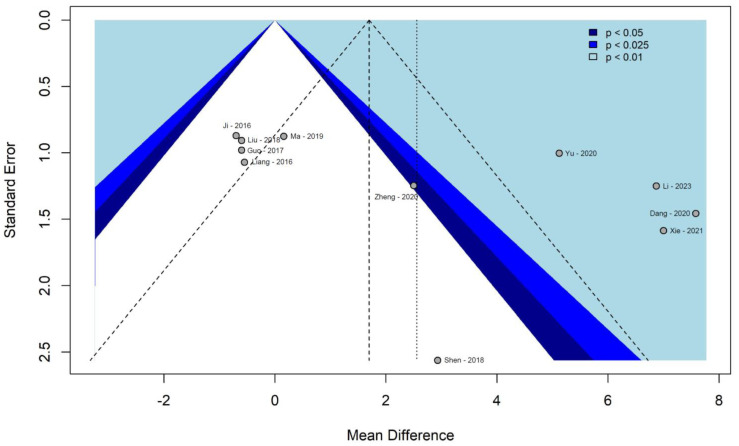
Primary endpoint—contour enhanced funnel plot of publication bias [38,39,40,41,42,43,44,45,46,47,48,49,50].

**Table 1 cancers-16-00199-t001:** Characteristics of included studies [38,39,40,41,42,43,44,45,46,47,48,49,50].

Study	Years of Enrollment	Study Design	Risk of Bias	RoB Tool Used	Total Pts	CME	Standard D2	Age D2	Age CME	Male Pts	Female Pts	Surgical Approach: Open (O) vs. Minimally Invasive (MI)	CME Type of Gastrectomy	D2 Type of Gastrectomy	TNM Stage CME	TNM Stage D2
Li—2015 [38]	2006–2011	Retrospective—random selection	Moderate	RoB 2	120	60	60	52.4	51.7	67	53	100% MI	Distal Gastrectomy 100%	Distal Gastrectomy 100%		
Ji—2016 [39]	2013–2015	Retrospective	Serious	ROBINS-I	98	48	50			55	43	(not specified)	Total Gastrectomy 100%	Total Gastrectomy 100%		
Liang—2016 [40]	2010–2013	Retrospective—random selection	Low	RoB 2	80	40	40	55.5	55.2	46	34	(not specified)	Total Gastrectomy 17,5%; Distal Gastrectomy 82,5%	Total Gastrectomy 22,5%; Distal Gastrectomy 77,5%	II 65%; III 35%	II 70%; III 30%
Guo—2017 [41]	2011–2014	Retrospective	Moderate	RoB 2	100	50	50	57.3	57.5	56	44	100% MI	Total Gastrectomy 100%	Total Gastrectomy 100%	II 58%; III 42%	II 64%; III 36%
Liu—2018 [42]	2016–2017	Retrospective—random selection	Moderate	ROBINS-I	84	42	42	54.6	54.3	42	42	(not specified)	Total Gastrectomy 100%	Total Gastrectomy 100%		
Shen—2018 [43]	2014–2017	Retrospective	Serious	ROBINS-I	85	45	40	63.2	62	54	31	100% MI	Distal gastrectomy 100%	Distal gastrectomy 100%	I 42.22% II 31.11% III 26.67%	I 47.5% II 12.5% III 40%
Luo—2018 [44]	2013–2015	Retrospective—random selection	Serious	ROBINS-I	66	33	33	57.4	57.3	41	25	100% MI for D2 type CME not specified	Total Gastrectomy 100%	Total Gastrectomy 100%		
Ma—2019 [45]	2014–2016	Retrospective—random selection	Low	RoB 2	96	48	48	61.6	61.5	54	42	(not specified)	Total Gastrectomy 100%	Total Gastrectomy 100%	I 43.7%; II 29.2%; II 27.1%	I 41.7%; II 33.3%; II 25%
Dang—2020 [46]	2018–2019	Retrospective—random selection	Low	RoB 2	98	49	49	51.4	49,8	70	28	100% MI	(not specified)	(not specified)	II 48.9%; III 51.1%	II: 44.9%; III: 55.1%
Yu—2020 [47]	2012–2017	Retrospective—random selection	Low	RoB 2	80	40	40	49.12	49.67	45	35	100% O	Total Gastrectomy 100%	Total Gastrectomy 100%		
Zheng—2020 [48]	2015–2017	Retrospective	Serious	ROBINS-I	165	84	81	63.0	63.1	137	28	100% MI	Total Gastrectomy 100%	Total Gastrectomy 100%	IB 10.71% IIA 30.95% IIB 15.47% IIIA 21.43% IIIB 15.47% IIIC 5.95%	IB 9.88% IIA 28.4% IIB 19.75 % IIIA 18.5% 18.5% 4.94%
Xie—2021 [49]	2014–2018	RCT	Low	RoB 2	338	169	169	54.5	54.8	213	125	100% MI	Distal gastrectomy 100%	Distal gastrectomy 100%	IB 20.7% IIA 29.6% IIB 15.4% IIIA 14.2% IIIB 18.9% IIIC 1.2%	IB 18.3% IIA 23.1% IIB 13.6% IIIA 20.1% IIIB 20.7% IIIC 4.2%
Li—2023 [50]	2014–2019	Retrospective	Moderate	ROBINS-I	599	367	232	65	63.7	434	165	100% MI	80.93% total gastrectomy; 18.8% distal gastrectomy; 0,.7% proximal gastrectomy	78.88% total gastrectomy; 19.4% distal gastrectomy; 1.72% proximal gastrectomy	I 10.89% II 36.78% III 52.32%	I 13.36% II 34.1% III 52.59%

**Table 2 cancers-16-00199-t002:** Subgroup analysis and metaregressions.

**Subgroup Analysis**								
**Variable**	**Number of Studies**	**MD**	**95% CI**		**I^2^ (%)**	***p* (Q Test—between Groups Differences)**		
			**Lower**	**Upper**				
Study design						0.04		
Retrospective	5	2.07	−1.91	6.04	86.76			
Retrospective—random selection	5	2.23	−2.38	6.84	90.42			
RCT	1	7.00	3.89	10.11	-			
Surgical approach						0.65		
Open	1	5.12	3.15	7.08	-			
Minimally invasive	6	4.35	0.83	7.88	86.87			
Type of gastrectomy (total gastrectomy)						0.039		
Distal	2	5.87	3.23	8.51	45.16			
Total	6	0.92	−1.54	3.39	82.33			
Risk of bias						0.458		
Low	5	3.73	−1.01	8.47	90.47			
Moderate	3	1.82	−8.82	12.45	92.81			
Serious	3	1.14	−3.93	6.22	62.75			
**Meta Regression Analysis**								
**Variable**	**Number of Studies**	**Estimate**	**95% CI**		** *p* **	**I^2^ (%)**	** *p* ** **(Q Test)**	**R^2^ (%)**
			**Lower**	**Upper**				
Year of publication	11	1.31	0.69	1.93	0.001	62.89	0.0039	76.87
Years of enrollment	11	1.08	−0.53	2.71	0.16	84.54	<0.0001	24.85
Total number of patients	11	0.012	−0.0007	0.025	0.062	83.63	<0.0001	32.1

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
