# Peer review of "Short-Term Outcomes after D2 Gastrectomy with Complete Mesogastric Excision in Patients with Locally Advanced Gastric Cancer: A Systematic Review and Meta-Analysis of High-Quality Studies"

_cancers, 2023, doi:10.3390/cancers16010199_

Round 1

Reviewer 1 Report

Comments and Suggestions for Authors

The authors have prepared a manuscript, “Short-term outcomes after D2 gastrectomy with complete mesogastric excision in patients with locally advanced gastric cancer: a systematic review and meta-analysis of high-quality studies.” The authors aimed to analyze the short-term outcomes of D2 radical gastrectomy with complete mesogastric excision (CME) compared to standard radical D2 gastrectomy using a systematic review. The primary outcome was the number of harvested lymph nodes expressed as mean difference and 95% confidence intervals. The authors included 13 studies for a total of 2019 patients and found that the mean number of harvested lymph nodes was significantly higher in the CME group. The authors also noted that the CME group had significantly lower intraoperative blood loss, lower length of stay and shorter operative times. The authors noted that the studies were burdened by serious risk of bias. The authors also concluded that radical gastrectomy with CME may provide a safe and more radical lymphadenectomy.

Comments:

Overall, the manuscript would benefit from minor English editing to improve readability.

Introduction:

-        In paragraph three of the introduction, the authors describe the procedure of CME in detail. There are no citations provided. If this procedure has been described in detail elsewhere with illustrations, it might make sense to include a citation or two. If it has not, I would recommend including an illustration.

-        D2 radical gastrectomy with CME and a standard radical D2 gastrectomy should be clearly described.

Methods:

-        The company name and location for “Rayyan web application,” or a URL should be provided.

-        The authors should clearly state in the inclusion/exclusion criteria whether they included studies described D2 radical gastrectomy with CME, studies describing standard D2 radical gastrectomy or ONLY studies that compared the two.

-        It appears to me that only studies that compare the two were included. This should be justified.

Results:

-        In the first sentence of the results, the authors state, “After the literature search, 13 high-quality studies were included…” This is prior to the assessment of study quality, in which the authors assert there was serious risk of bias.

-        What does “canalization” refer to?

Discussion:

-        The authors report that the mean number of LNs harvested was significantly higher among patients receiving CME. Is this a clinically significant difference when the means for both CME and standard D2 are so high?

-        The authors report significant differences in secondary outcomes as well – are these clinically significant differences?

-        How do the authors explain the differences in secondary outcomes between D2 with CME and standard D2?

-        The authors should spend more time discussing the limitations of this study.

-        Page 12 – Line 351 – Please provide the references where you indicate (ref)

Comments on the Quality of English Language

Moderate English editing of the manuscript could improve readability. 

Author Response

Please find attached the replies to reviewers

Reviewer 2 Report

Comments and Suggestions for Authors

In general, it is a well-structured work. I miss two important points:

-the limitations of the study.

-More conclusions. With all the data collected, is it not possible to conclude more aspects?

Author Response

(The authors gave the same response as above.)

Round 2

Reviewer 1 Report

Comments and Suggestions for Authors

The authors have reasonably responded to reviewers concerns.